# Liver Elastography Methods for Diagnosis of De Novo and Recurrent Hepatocellular Carcinoma

**DOI:** 10.3390/diagnostics15091087

**Published:** 2025-04-25

**Authors:** Razvan Cerban, Speranta Iacob, Carmen Ester, Mihaela Ghioca, Mirela Chitul, Razvan Iacob, Liana Gheorghe

**Affiliations:** 1Faculty of Medicine, Carol Davila University of Medicine and Pharmacy, 050474 Bucharest, Romania; cerbanrazvan@yahoo.com (R.C.); carmen.ghidu@gmail.com (C.E.); mirela.onica@drd.umfcd.ro (M.C.); raziacob@gmail.com (R.I.); drlgheorghe@gmail.com (L.G.); 2Center for Digestive Diseases and Liver Transplant, Fundeni Clinical Institute, 022328 Bucharest, Romania; lita.mihaela.corina@gmail.com

**Keywords:** liver elastography, hepatocellular carcinoma (HCC), liver fibrosis assessment, acoustic radiation force impulse (ARFI), tumor stiffness, transient elastography (TE), noninvasive imaging, HCC risk prediction

## Abstract

Hepatocellular carcinoma (HCC), a common consequence of chronic liver disease, ranks among the most prevalent cancers globally and contributes significantly to cancer-related mortality. Liver fibrosis is intimately associated with hepatic function and the likelihood of future HCC occurrence. Despite the fact that liver biopsy continues to be the gold standard for diagnosing fibrosis, its utility is hindered by cost and invasiveness, along with patient unease, procedural rejection, and potential adverse effects. Liver elastography has become a leading noninvasive means of assessing tissue stiffness with considerable diagnostic precision. Malignant tumors generally exhibit higher cellularity in comparison to benign ones, resulting in increased stiffness. Elastography techniques capitalize on alterations in tissue elasticity stemming from specific pathological or physiological processes. Technological innovations, such as advanced ultrasound imaging and artificial intelligence (AI)-integrated systems, are paving the way for enhanced diagnostic accuracy and risk prediction. Recent research underscores the potential of elastography in managing HCC patients, presenting novel clinical applications, including prediction of HCC development, differentiation between malignant and benign liver lesions, evaluating treatment response, and forecasting recurrence post-treatment, though certain findings remain contentious. Therefore, this review aims to sum up the latest advancements in liver elastography for HCC patients, outlining its applications while addressing existing limitations and avenues for future progress.

## 1. Introduction

Today, chronic liver diseases represent an important global health burden, with a great impact on morbidity and mortality. Common causes include chronic hepatitis B virus (HBV) and hepatitis C virus (HCV) infections, alcohol abuse, and metabolic dysfunction-associated steatotic liver disease (MASLD) [1]. These conditions can lead to liver fibrosis and ultimately cirrhosis, which can result in complications, including increased portal pressure, hepatic insufficiency, or progression to hepatocellular carcinoma (HCC) [2].

Liver fibrosis assessments traditionally relied on liver biopsy, which was considered the gold standard to assess fibrosis severity and evaluate necroinflammatory activity, using various semiquantitative scoring models, like the METAVIR score, for chronic HBV or HCV infections. Despite its historical significance, reliance on biopsy has decreased due to the growing recognition of its limitations and the availability of less invasive techniques. Most fibrosis classification systems, apart from the Ishak score, follow an F0–F4 scale, with F0 indicating no fibrosis, F1 denoting mild fibrosis, F2 signifying moderate fibrosis, F3 reflecting advanced fibrosis, and F4 representing cirrhosis [3,4]. However, liver biopsy has drawbacks, including invasiveness and associated complications, like pain or bleeding, thus limiting patient acceptability and feasibility for repeated assessments and ongoing disease monitoring. Additionally, since a biopsy examines only limited liver tissue, it may lead to variability in sampling and possible diagnostic inaccuracies [5]. Variability among observers further complicates interpretation [6]. Moreover, liver biopsy lacks dynamic information on disease progression, making it an imperfect reference standard. This necessitates a shift toward more patient-friendly, dynamic, and precise diagnostic tools that can assess liver health comprehensively.

Novel, noninvasive approaches to assess liver health have been devised, including serum markers, ultrasound (US) elastography, and magnetic resonance (MR) imaging. The integration of these methods into clinical practice has revolutionized chronic liver disease management by providing reliable diagnostic and prognostic data without invasive procedures.

Serum markers encompass both simple indicators, such as number of thrombocytes, aspartate aminotransferase-to-platelet ratio index (APRI) [7], and FIB-4 [8], as well as more intricate, proprietary composite scores, like FibroTest (BioPredictive, Paris, France), available at https://www.biopredictive.com, accessed on 11 April 2025 [9] and FibroMeter (Echosens, Paris, France), available at https://www.fibrometer.com, accessed on 11 April 2025 [10]. While these tests are relatively straightforward to administer, they demonstrate limited accuracy, particularly in intermediate fibrosis stages [11], and are generally deemed less precise compared to elastographic techniques [12]. Additionally, serum biomarkers may be influenced by extrahepatic factors, further limiting their reliability in certain populations.

The severity of hepatic fibrosis appears strongly linked to the risk of developing HCC. Moreover, more than 70% of HCC cases arise in the setting of advanced fibrosis or previously established liver cirrhosis [13]. Hepatic fibrosis is also closely associated with liver functional reserve, a critical factor in devising treatment strategies and determining prognosis in HCC patients. Thus, precise staging of fibrosis is pivotal in stratifying HCC risk and optimizing clinical management.

## 2. Elastography Overview

Elastography, initially conceptualized by Ophir, pertains to the noninvasive evaluation of tissue mechanical characteristics, particularly elasticity, representing tissue’s resistance to deformation under stress [14]. Since its inception, elastography has undergone significant advancements, evolving into a robust diagnostic modality.

In elastography-based quantification, shear-wave propagation facilitates stress generation, occurring either transiently from a single mechanical trigger, or dynamically through sustained acoustic wave application. Malignant tumors often exhibit increased cellularity, which, alongside extracellular matrix alterations and stromal remodeling, can contribute to greater tissue stiffness. However, the relationship between cellularity and mechanical stiffness is complex and influenced by additional factors, such as fibrosis, necrosis, and tumor heterogeneity. Consequently, tumor stiffness emerges as a potential imaging biomarker for distinguishing tumor characteristics, but it should be considered a complementary tool, not a stand-alone diagnostic criterion. This provides an opportunity for personalized care, as stiffness values can guide treatment decisions and monitor therapeutic response.

Elastography utilizes an applied force, combined with a detection system, to evaluate tissue deformation. The forces used in this technique can be categorized as follows:Static compression elastography (strain elastography) evaluates tissue stiffness by applying manual or physiological compression (natural organ activity, such as cardiac motion, vascular pulsations, and respiratory movements) and measuring tissue deformation. Unlike shear wave-based methods, it provides relative stiffness rather than absolute stiffness values [15]. Although strain elastography is not recommended for liver fibrosis staging due to its operator dependency and lack of standardization, its potential role in the evaluation of focal liver lesions, particularly hepatocellular carcinoma (HCC), has been explored in several studies.Dynamic compression, which involves sustained oscillations at a fixed frequency.Impulse compression, where brief vibrations result from either an external mechanical impulse (e.g., in FibroScan) or an ultrasound-induced impulse (as in ARFI and SWE), both of which generate shear waves.

While contrast-enhanced imaging remains the primary diagnostic approach for hepatocellular carcinoma (HCC), recent reports indicate that noninvasively measured hepatic nodule stiffness could aid in distinguishing between malignant and benign liver lesions.

Quantitative elastography approaches encompass techniques like transient elastography (TE) and different ARFI-based methods, including two-dimensional shear-wave elastography (SWE) and point shear-wave elastography (pSWE).

### 2.1. Transient Elastography

The initial commercially available TE system, the FibroScan from Echosens, was launched in 2003, in Europe, and was approved by the Food and Drug Administration (FDA) in the United States in 2013. This system utilizes a 50 Hz mechanical impulse externally on the skin, followed by the measurement of the shear-wave velocity. A range of probes are available, with the M probe primarily utilized for routine evaluations, whereas the XL probe enhances precision in patients with higher body mass [16]. Unlike the M probe, the XL probe measures deeper tissue layers (35–75 mm vs. 25–65 mm) and functions at a lower frequency (2.5 MHz vs. 3.5 MHz). ARFI elastography employs targeted ultrasound pulses to induce tissue displacement, leading to the propagation of shear waves for stiffness assessment [17].

### 2.2. Acoustic Radiation Force Impulse (ARFI)-Based Elastography Methods

ARFI (acoustic radiation force impulse) elastography is a noninvasive imaging technique used to assess tissue stiffness and works by applying short, focused ultrasonic pulses to tissue, generating localized displacements.

The terminology associated with ARFI elastography in the scientific literature remains inconsistent. Although both pSWE and 2D SWE utilize ARFI to generate shear waves, current research often refers to pSWE specifically as ARFI elastography.

Siemens (pSWE, Virtual Touch Quantification) and Supersonic Imagine (2D SWE) were among the first to introduce ARFI techniques in clinical applications. Today, these methods have been adopted by other leading manufacturers, including Philips [18], Hitachi [19], GE [20], Toshiba [21], and Samsung [22].

Clinical ultrasound elastography systems typically present “stiffness” results in either Young’s modulus (E, measured in kilopascals), shear-wave propagation speed (in meters per second), or a combination of both. Assuming tissue is incompressible, the equation E = 3ρc^2^ establishes the connection between shear-wave velocity (c) and Young’s modulus (E), where ρ signifies the assumed density of the tissue, generally approximated to water’s density. According to the 2017 guidelines from the European Federation of Societies for Ultrasound in Medicine and Biology (EFSUMB), it is recommended to perform pSWE and 2D-SWE assessments more than 1 cm below the liver capsule to ensure accuracy and optimal results [23].

A challenge in directly comparing these methods is the impact of frequency on biological tissue properties. Increased shear-wave frequency results in greater stress and strain rates, which correspond to elevated stiffness measurements [24]. This disparity can pose challenges when evaluating US elastographic techniques, as transient elastography (TE) operates at 50 Hz, while acoustic radiation force impulse (ARFI) methods operate at frequencies ranging from 100 to 500 Hz [25]. When analyzing elastography methods, it is essential to account for frequency dependence; the measurement technique used; and the parameter being reported, such as wave velocity, Young’s modulus (E), or shear modulus (G).

Clinically available ultrasound (US) techniques are detailed in Table 1.

## 3. Clinical Applications of Liver Elastography

### 3.1. Prediction of Hepatocellular Carcinoma (HCC) Development and Portal Hypertension (PHT) Occurrence

Estimating the probability of HCC development in those diagnosed with chronic liver disease (CLD) is a key aspect of clinical care. Fibrosis of the liver serves as a highly predictive indicator, reflecting hepatic function and the likelihood of progression to HCC [26]. Emerging data suggest that combining elastographic findings with genetic and molecular markers could further enhance risk prediction, paving the way for tailored surveillance programs.

The combination of elastographic measurements with genetic risk scores has shown promise in improving predictive models for HCC [27]. By leveraging these multifaceted approaches, healthcare providers can more effectively identify high-risk patients and implement personalized monitoring and intervention strategies, ultimately improving clinical outcomes for those at risk of HCC and portal hypertension.

A study examining longitudinal variations in liver stiffness values revealed that patients experiencing an increase in liver stiffness between 1 and 1.5 kPa/year faced an estimated tenfold rise in complications [28]. Furthermore, a meta-analysis assessing hepatic complications demonstrated that each additional unit of liver stiffness was linked to a 7% rise in the probability of decompensation [29].

Transient elastography (TE) also aids in diagnosing an increase in portal pressure, with a meta-analysis highlighting its robust diagnostic performance in identifying clinically significant portal hypertension (characterized by a hepatic venous pressure gradient exceeding 10 mm Hg) [30].

Additionally, scores integrating liver stiffness associated with number of platelets and spleen size at ultrasound, or portal hypertension risk scores, have been devised to enhance accuracy of diagnosis [31]. A sizable prospective study from Japan with 866 patients reported a significantly elevated incidence of HCC over a three-year period among individuals whose initial liver stiffness exceeded 25 kPa, relative to those with values below 10 kPa [32]. Preliminary studies evaluating point shear-wave elastography (pSWE) for predicting HCC development suggest that hepatic stiffness measurements determined by using pSWE are a noteworthy predictive parameter, although further validation is warranted [33].

Direct-acting antivirals (DAAs) have significantly improved HCV natural history, leading to high cure rates and reduced liver-transplant needs. However, some patients still develop HCC post-sustained virological response (SVR), highlighting the importance of fibrosis assessment for prognosis and monitoring. Research indicates liver stiffness can decrease by up to 35% after DAA therapy, as measured by TE [34,35] and acoustic radiation force impulse (ARFI) methods [36]. Ongoing surveillance with elastography allows for early detection of stiffness increases, prompting further investigation and potentially influencing HCC screening strategies.

However, the clinical implications of decreased liver stiffness post-sustained virological response remain uncertain, given the limited longitudinal studies including biopsy compared with noninvasive tests. One study examining changes over time in liver stiffness with paired liver biopsies in HCV/HIV-coinfected individuals undergoing antiretroviral therapy associated with anti-HCV treatment revealed that patients with progressing fibrosis exhibited an increase in hepatic stiffness at three years after initial assessment, whereas hepatic stiffness remained unchanged or decreased in stable patients [37].

Numerous longitudinal prospective studies have reported a relationship between TE-measured liver stiffness and the likelihood of HCC development [32,38,39,40,41,42,43,44,45,46,47,48,49]. Findings on the correlation between liver stiffness measurements and clinically significant portal hypertension, including validated cutoff values and predictive models, are summarized in Table 2. However, the predictive accuracy of these measurements for outcomes like HCC risk is influenced by liver disease etiology and treatment status. Therefore, etiology-specific algorithms are needed for accurate risk stratification. Integrating these factors into AI-driven predictive models could refine HCC risk assessment by weighting etiology and treatment response alongside elastography values.

Current evidence supports the use of elastographic techniques, particularly TE and SWE, as valuable tools for assessing portal hypertension and predicting liver-related outcomes, including hepatic decompensation and HCC. While liver stiffness measurement (LSM) remains central to fibrosis staging and longitudinal monitoring, spleen stiffness measurement (SSM) has emerged as a superior noninvasive surrogate for CSPH. Several studies have demonstrated that spleen stiffness values > 46–50 kPa correlate with CSPH, and values above 55–60 kPa may identify patients at higher risk for varices requiring treatment [50,51]. Consequently, the Baveno VII criteria [52] have incorporated SSM alongside LSM and platelet count to noninvasively rule out high-risk varices and reduce unnecessary endoscopies. Moreover, recent data suggest that SSM is less affected by antiviral-induced reductions in hepatic inflammation, making it a potentially more stable and reliable longitudinal marker of residual portal hypertension and HCC risk after SVR. The study by Dajti et al. [53] demonstrated that SSM at 6 months post-SVR was an independent predictor of HCC development, particularly in patients with LSM-SVR24 > 10 kPa, where only those with SSM > 42 kPa had significantly elevated HCC risk, highlighting that persistent portal hypertension, rather than fibrosis alone, drives post-SVR carcinogenesis. This led to a proposed algorithm stratifying patients into low-, moderate-, and high-risk HCC categories based on combined LSM and SSM values, allowing for personalized surveillance strategies. These findings underscore the importance of integrating both liver and spleen stiffness into clinical algorithms and AI-based predictive models, which can dynamically adjust risk assessment based on etiology, treatment response, and portal pressure surrogates.

### 3.2. Discriminating Between Benign and Malignant Liver Lesions, as Well as Characterization of HCC

There are existing data evaluating the role of ultrasound elasticity-based techniques in measuring tumor stiffness for liver lesion assessment [54,55,56,57,58,59,60,61,62] with a tendency toward higher stiffness in malignant tumors, like HCC. Furthermore, elastographic techniques such as SWE and pSWE have shown promise in differentiating benign from malignant focal liver lesions based on stiffness thresholds [63,64]. While threshold values vary, likely due to differences in equipment and patient populations, the overall trend supports the diagnostic accuracy of elastography. Notably, point SWE (pSWE) shows particular promise in improving differentiation.

Our group was among the first to report the utility of real-time strain elastography (RTE) in the noninvasive characterization of small HCC nodules in cirrhotic patients, showing excellent diagnostic performance, especially when combined with Doppler ultrasound. In a prospective study of 42 patients with 58 liver nodules (1–3 cm), we found that color-coded strain patterns that specifically increased blue color intensity were independently predictive of HCC, with a diagnostic accuracy (AUROC) of 0.94 [65]. More recent developments, such as improved convex probes and semiquantitative analysis, have renewed interest in RTE applications in liver pathology, including tumor characterization and treatment monitoring. In this context, RTE has been successfully used to assess the effects of radiofrequency ablation (RFA) in experimental models, showing comparable accuracy to shear-wave elastography (SWE) and histological measurements [66,67]. While these applications remain limited to specific clinical scenarios, they underline the potential complementary role of elastographic techniques beyond fibrosis assessment.

In essence, ultrasound elastography offers a noninvasive tool for improving the characterization of focal liver lesions. It has the potential to reduce the need for invasive procedures and guide clinical decision-making in HCC diagnosis and treatment planning. However, further research is still required to standardize cutoff thresholds, optimize acquisition protocols and integrate elastographic data into multiparametric diagnostic algorithms.

Table 3 summarizes the findings from various studies evaluating the accuracy of shear-wave elastography (SWE) and acoustic radiation force impulse (ARFI) elastography in differentiating benign from malignant focal liver lesions (FLLs). The table includes threshold values and key diagnostic performance metrics, highlighting the sensitivity and specificity of each method.

### 3.3. Prediction of Post-Treatment Complications

Liver resection stands out as the most effective and readily accessible therapy for patients diagnosed with early-stage HCC. However, the majority of HCC patients present with advanced liver fibrosis or cirrhosis, significantly increasing the risk of severe postoperative complications, particularly in cases requiring major hepatectomy. Postoperative liver failure and decompensation remain major concerns. Recent studies have highlighted the utility of liver stiffness measurement (LSM) by using transient elastography (TE) and other elastographic techniques in predicting postoperative outcomes. This highlights the growing role of elastography in the perioperative setting, particularly for patients with borderline liver function or complex surgical needs.

A comparative study evaluated LSM against the hepatic venous pressure gradient (HVPG) and demonstrated that LSM effectively predicts decompensation risks at three months post-surgery in cirrhotic patients undergoing liver resection [39]. High LSM values over 16.2 kPa correlate with reduced hepatic functional reserve and increased likelihood of postoperative complications, including hepatic insufficiency and ascites [68]. This correlation underscores the potential of LSM to serve as a reliable surrogate marker for hepatic venous pressure, offering a noninvasive alternative to invasive HVPG measurement.

Moreover, combining LSM with other clinical parameters, such as platelet count and spleen stiffness, can enhance predictive accuracy for postoperative outcomes [31]. Patients with lower preoperative LSM values have been shown to have better recovery profiles, making elastography an invaluable preoperative risk stratification tool. Integrating elastographic data into multidisciplinary surgical planning could reduce complication rates and improve long-term outcomes in HCC patients.

In a study comparing liver stiffness measurement (LSM) by TE with hepatic venous pressure gradient (HVPG) for predicting decompensation after surgical resection in patients with cirrhosis and early HCC, TE demonstrated good predictive ability for decompensation at 3 months post-surgery [69]. These findings support the integration of noninvasive elastographic techniques into preoperative assessment protocols, potentially reducing the burden of complications and improving surgical outcomes for HCC patients. Results from studies investigating the role of liver stiffness measurement in predicting complications such as liver failure, decompensation, and post-surgical outcomes in HCC patients are presented in Table 4.

### 3.4. Biomarker of Treatment Response

Early recurrence of tumors post-treatment signifies a grim prognosis for patients with hepatocellular carcinoma (HCC). Hence, integrating risk assessment for tumor recurrence into guiding therapeutic strategies is imperative. Earlier reported research has pinpointed that microvascular invasion and poorly differentiated tumors are major predictors of HCC recurrence following liver transplantation or surgical removal. Moreover, various studies have highlighted that in patients with viral hepatitis, liver stiffness measurement (LSM) by Fibroscan has been shown to independently predict the likelihood of HCC recurrence post-treatment [71,72].

Overall, these studies illustrate the clinical utility of LSM as a noninvasive biomarker for predicting treatment response and disease recurrence in HCC. By integrating elastography into routine clinical assessments, clinicians can enhance treatment decision-making, optimize patient stratification, and improve long-term outcomes.

Results from studies investigating the significance of liver stiffness assessment in predicting recurrence after HCC treatment are presented in Table 5.

This proposed table (Table 6) outlines a structured approach to the use of liver stiffness measurement (LSM) in the management of chronic liver disease.

### 3.5. Future Directions: Elastography and AI Integration

Artificial intelligence (AI) is set to revolutionize elastography-based liver assessment by enhancing image acquisition, interpretation, and clinical decision-making. Traditional elastography methods rely on manual region of interest (ROI) selection and operator-dependent data acquisition, introducing variability. AI-driven tools could automate these processes, reducing human error and improving reproducibility. ROI selection significantly impacts measurement accuracy, requiring operators to manually position ROIs while avoiding vessels and bile ducts. AI algorithms trained on large elastography datasets could automate this process, ensuring consistent and standardized placement across patients and imaging sessions. Elastography is susceptible to motion artifacts from respiration, cardiac pulsation, and probe pressure. AI-powered real-time motion correction can filter out transient fluctuations and improve signal stability. Deep learning models analyzing raw elastography waveforms could refine shear wave-propagation mapping, enhancing liver stiffness measurement accuracy.

Beyond image optimization, AI could refine hepatocellular carcinoma (HCC) risk prediction by integrating elastography data with clinical, laboratory, and genetic markers. Machine learning models analyzing longitudinal liver stiffness trends, platelets, AFP levels, and genetic predisposition could generate personalized HCC risk scores.

While elastography provides insights into liver stiffness and fibrosis progression, its diagnostic accuracy for HCC could be further enhanced by integrating emerging biomarkers, including proteomic, metabolomic, and circulating tumor DNA (ctDNA) analyses. This multimodal approach may improve early detection, particularly in patients without clear radiologic evidence of malignancy.

AI-enhanced liver stiffness measurement (LSM) trends may surpass alpha-fetoprotein (AFP) in predicting HCC risk, particularly in metabolic dysfunction-associated steatotic liver disease (MASLD), where AFP sensitivity is limited [73]. Furthermore, AI algorithms have been developed to predict hepatocellular carcinoma (HCC) risk by integrating LSM with clinical features [27]. The SMART-HCC score, a liver stiffness-based machine learning algorithm, demonstrated superior predictive accuracy for HCC risk stratification across various chronic liver diseases, outperforming existing risk scores. This model incorporates variables such as liver stiffness measurement (LSM), age, gender, etiology of liver disease, hypertension status, alanine aminotransferase (ALT), alkaline phosphatase (ALP), platelet count, and creatinine levels to assess an individual’s risk of developing HCC. An online calculator for the SMART-HCC score is available at https://tcfyip.shinyapps.io/smart_hcc_score/ (accessed on 11 April 2025), providing clinicians with a practical tool for risk stratification in patients with CLDs [74].

AI-driven shear-wave elastography (SWE) and acoustic radiation force impulse (ARFI) imaging could improve lesion characterization, reducing the need for liver biopsies while maintaining diagnostic accuracy. AI-assisted multiparametric ultrasound may outperform conventional ultrasound in early HCC detection, especially in cirrhotic livers, where echotextural analysis is challenging.

Despite its potential, AI integration into liver elastography faces challenges before widespread adoption. Standardization across ultrasound platforms, validation in diverse patient cohorts, and regulatory approval are critical for clinical implementation.

### 3.6. Best Practices for Liver Elastography in Clinical Use

Various technical and patient-related factors can influence measurement accuracy. To ensure reliability, clinicians should adhere to standardized protocols for patient preparation, measurement acquisition, and interpretation. Patient preparation is a key factor in obtaining accurate elastography results. Patients should fast for at least 3–4 h before undergoing elastography to minimize postprandial variations in liver stiffness, which may transiently increase measurements due to changes in hepatic blood flow [25]. Additionally, strenuous physical activity before the procedure should be avoided, as it can induce transient increases in liver stiffness. Proper patient positioning and probe selection are essential for optimizing results. For TE, the patient should be positioned supine with the right arm extended overhead to maximize intercostal space exposure, facilitating optimal probe placement [69].

Regarding probe selection, the M probe is recommended for most patients, whereas the XL probe is preferred for individuals with BMI ≥ 30–32 kg/m^2^ or when the skin-to-capsule distance is ≥25 mm [75]. The XL probe is designed to improve the feasibility and reliability of LSM in patients with increased subcutaneous tissue thickness, ensuring accurate assessments of liver fibrosis.

In SWE, probe selection depends on the patient’s body habitus and the system in use. Convex probes are typically used, with adjustments made based on liver depth and image quality.

In both TE and SWE, measurements should be obtained from homogeneous liver parenchyma, at a depth of 1–2 cm below the capsule, and avoiding large vessels. Region of interest (ROI) selection also plays a critical role in obtaining valid results. Measurements should be performed at least 1–2 cm below the liver capsule, as subcapsular fibrosis can lead to overestimated stiffness values. Additionally, the ROI should be placed away from large blood vessels and bile ducts, as these structures can interfere with shear-wave propagation and result in inaccurate readings [76]. In TE, stiffness is measured using a fixed-volume cylindrical ROI, while SWE provides real-time B-mode guidance, allowing visual placement of the ROI in areas with homogeneous tissue and minimal artifact. SWE elastograms should show consistent shear-wave propagation, and ROI placement should reflect stable, artifact-free areas of the liver.

Quality control and reproducibility are essential to ensure the reliability of elastography measurements. A minimum of 10 valid acquisitions should be obtained per session to reduce variability. For both TE and SWE, an interquartile range (IQR) to median ratio of ≤30% is recommended as a quality criterion to confirm measurement reliability. For SWE, color map stability and propagation consistency should also be visually assessed during acquisition. If variability between measurements is high, repositioning the probe and repeating acquisitions may improve accuracy. LSM values should at all times be interpreted in the clinical context, accounting for potential confounders such as inflammation, congestion, or cholestasis, which may elevate stiffness independently of fibrosis.

### 3.7. Assessment of Diagnostic Reliability and Technical Failure in Elastography

#### 3.7.1. Transient Elastography (TE) Technique

In a paper encompassing 13,369 assessments utilizing the M probe, the success rate and technical limitations of TE were evaluated [77]. The technique experienced failure in 3.1% of cases, with an additional 15.8% of cases yielding unreliable measurements. Notably, body mass index emerged as a key element contributing to either invalid or inconsistent measurements. As obesity continues to rise globally, addressing technical limitations in this population is crucial for widespread adoption of TE. The implementation of the XL probe has notably enhanced the accuracy of TE measurements, particularly in patients with metabolic-associated steatotic liver disease (MASLD) [78,79,80]. For instance, a study involving 276 subjects showcased that accurate readings with the XL probe were achieved in 73% of patients, in comparison to about 50% of patients when using the M probe [79].

#### 3.7.2. Shear-Wave Elastography Techniques, Including pSWE and 2D SWE

The reliability of both pSWE and 2D SWE was evaluated in a cohort of 79 subjects, with evaluations conducted by three radiology doctors [81]. The unsuccessful measurement rates were low for both methods, i.e., under 1% for pSWE and approximately 5% for 2D SWE, and intraobserver consensus was higher for pSWE compared to 2D SWE (0.915 vs. 0.829). A second scanning took place using 2D-SWE measurements, performed by the same operator, on the same day, as the first scan demonstrated excellent results, with intraclass correlation coefficients (ICCs) of 0.95 and 0.93 for a skilled and trainee operator, respectively. These findings emphasize the critical role of operator training and standardization to ensure consistent, high-quality results across clinical settings. Nevertheless, intraobserver consistency between repeat assessments in the same patient on separate days showed ICC values of 0.84 for an experienced operator and 0.65 for a beginner [82]. Evidence suggests that operator experience influences pSWE measurements, underscoring the importance of adequate operator training. As pSWE and 2D SWE are adopted by different ultrasound system manufacturers, potential discrepancies between platforms should be carefully considered.

#### 3.7.3. Diagnostic Limitations of Ultrasound-Based Elastography

A single probe can be used for both 2D SWE and pSWE across all patients, regardless of body mass, as the region of interest can be placed at varying depths within the liver. Unlike transient elastography (TE), the presence of ascites does not pose a limitation for ARFI ultrasound methods, allowing its application even in patients with decompensated liver cirrhosis for prognostic purposes. However, there is a reported potential for overestimation of liver stiffness values, with additional confounding factors, like liver enzyme flares [83,84] or acute viral hepatitis [85], excessive alcohol intake [86,87], and congestive heart failure [88]. Developing standardized protocols to adjust for these confounders remains a priority in refining elastographic accuracy.

Efforts have been made to establish cutoff values that accommodate these interfering factors, though additional validation is necessary. Steatosis continues to be a subject of discussion, as research findings vary; certain studies suggest it has a negative impact [89], while other studies find no significant impact [90].

In conclusion, the utilization of ultrasound elastographic techniques necessitates adherence to a standardized protocol and the critical interpretation of results, considering the influence of confounding factors.

Despite its widespread clinical use, liver elastography lacks universal standardization across different platforms. Various elastography techniques, including pSWE and 2D SWE, are available from multiple manufacturers, each using proprietary algorithms, pulse frequencies, and reporting parameters. This variability complicates direct comparison of liver stiffness values and may lead to misinterpretation when transitioning between devices or incorporating results from different studies. One major limitation is the difference in shear-wave frequencies used across techniques, as already mentioned. As shear-wave velocity is frequency-dependent, these differences result in significant variability in stiffness values, even within the same patient [90]. Furthermore, measurement depths, acquisition angles, and shear-wave dispersion methods differ between vendors, further complicating standardization. The absence of universal cutoff values for fibrosis staging presents an additional challenge, as clinicians may struggle to interpret elastography results consistently across different systems.

To improve reproducibility and clinical applicability, multicenter studies should be conducted to harmonize liver stiffness measurement thresholds across different techniques. The development of standardized conversion formulas or correction factors between TE, pSWE, and 2D SWE would facilitate cross-platform comparison and improve diagnostic consistency [91]. Regulatory and professional bodies such as the European Federation of Societies for Ultrasound in Medicine and Biology (EFSUMB), the World Federation for Ultrasound in Medicine and Biology (WFUMB), and the Radiological Society of North America (RSNA) should collaborate to establish standardized acquisition protocols and validation frameworks. Emerging technologies, such as AI-based algorithms, could facilitate cross-vendor calibration by identifying correction factors between different elastography devices. Machine learning models trained on multi-platform datasets may help reduce discrepancies and improve the reproducibility of liver stiffness measurements across clinical settings.

## 4. Conclusions

As a key advancement, liver elastography is widely utilized in present-day diagnosis and treatment strategies in patients with chronic liver diseases. By offering a noninvasive alternative to liver biopsy, elastography not only improves patient comfort but also provides dynamic insights into disease progression, risk stratification, and therapeutic monitoring. While elastography has already transformed the landscape of liver disease evaluation, its true potential lies ahead. The integration of artificial intelligence promises to refine measurements, enhance diagnostic accuracy, and provide real-time clinical insights. Moreover, combining elastography with other imaging modalities, such as contrast-enhanced ultrasound, could unlock new opportunities for precision diagnostics and personalized treatment strategies. To fully realize its potential, the standardization of protocols and large-scale validation studies are essential. Collaboration among clinicians, researchers, and the medical technology industry will accelerate innovation, paving the way for elastography to become a cornerstone of personalized medicine.

In conclusion, liver elastography is not merely a diagnostic tool; it represents a paradigm shift in hepatology. With its ability to provide comprehensive liver assessments and predict patient outcomes, elastography holds the promise of shaping the future of liver disease management and elevating standards of care worldwide.

## Figures and Tables

**Table 1 diagnostics-15-01087-t001:** Overview of different elastography techniques.

Technique	Ultrasound Frequency (MHz)	Availability	Cost	Liver- Sampling Volume	Region of Interest	Cause of Measurement Inaccuracy	Evidence
Transient elastography	2.5–3.5 MHz	Widespread	Low	Small	Restricted, no guidance	Ascites, obese patients (M probe)	Excellent validation
Point shear-wave elastography (pSWE)	4–9 MHz for deep tissue	Moderate	Low	Small	Flexible with US guidance; recommended 1 cm below liver capsule and 5 cm from transducer	High body mass-index standardization needed	Moderate, good validation
2D shear-wave elastography (2D SWE)	9–15 MHz for superficial tissue	Limited	Low	Medium	Flexible with US guidance	High body mass-index standardization needed	Moderate, good validation

**Table 2 diagnostics-15-01087-t002:** Elastography-based prediction of portal hypertension.

Reference Number	Source Citation	Number of Patients	Method	Country/Region	Threshold Values	Diagnostic Performance	Key Findings
[26]	Taylor et al., *Gastroenterology*, 2020	11,000	TE	Multiple countries	Varies by fibrosis stage	Prognostic accuracy for NAFLD outcomes	Liver stiffness correlates with fibrosis severity and clinical outcomes in NAFLD.
[27]	Chan et al., *JNCI*, 2024	10,000	SWE	Multiple countries	Liver stiffness-based risk score	HCC prediction accuracy	Liver elastography-based risk score predicts HCC risk effectively.
[28]	Corpechot et al., *Gastroenterology*, 2014	180	TE	France	10–20 kPa for severe fibrosis	Predictive accuracy for PSC progression	Baseline and changes in liver stiffness are associated with fibrosis severity and outcomes in PSC.
[29]	Singh et al., Clin *Gastroenterol Hepatol*, 2013	16,000	TE	Multiple countries	Fibrosis staging thresholds	Prognostic value for decompensation and mortality	Liver stiffness is linked to decompensation, HCC, and death
[30]	Shi et al., *Liver Int*, 2013	1300	TE	Multiple countries	>13.6 kPa for PH	Sensitivity: 88% Specificity: 86%	TE accurately evaluates portal hypertension in chronic liver disease.
[31]	Berzigotti et al., *Gastroenterology*, 2013	150	TE	Italy	>21 kPa for PH	Combined with spleen size and platelets for accuracy	TE combined with clinical parameters identifies portal hypertension in compensated cirrhosis.
[32]	Masuzaki et al., *Hepatology*, 2009	492	TE	Japan	>25 kPa for HCC risk	Sensitivity: 81% Specificity: 77%	TE predicts HCC development in hepatitis C patients.
[33]	Hernandez Sampere et al., *Hepatol Commun*, 2020	370	pSWE	Germany	>9.5 kPa for fibrosis	Liver-related event prediction	ARFI demonstrated predictive value for liver complications associated with chronic viral hepatitis.
[34]	Elsharkawy et al., *J Gastroenterol Hepatol*, 2017	302	TE	Egypt	Post-treatment fibrosis reduction	Change in stiffness post-SVR	TE shows fibrosis improvement after antiviral therapy.
[35]	Sáez-Royuela et al., *Eur J Gastroenterol Hepatol*, 2016	128	TE	Spain	Post-treatment stiffness thresholds	TE reduction post-HCV therapy	TE effectively assesses fibrosis regression post-therapy.
[36]	Suda et al., *World J Hepatol*, 2017	282	SWE	Japan	TE vs. SWE thresholds	Pre- vs. post-treatment fibrosis assessment	SWE accurately tracks fibrosis changes in HCV patients after antiviral therapy.
[37]	Schmid et al., *PLoS One*, 2015	165	TE	Switzerland	Staging thresholds	Comparison with biopsy	TE provides comparable results to biopsy in HIV/HCV coinfected patients.
[38]	Kim et al., *PLoS One*, 2012	578	TE	South Korea	>12 kPa for fibrosis	Prediction of liver-related events	TE predicts liver-related complications in HBV patients.
[39]	Robic et al., *J Hepatol*, 2011	121	TE	France	>13.6 kPa for PH	PH-related complications prediction	TE accurately predicts portal hypertension complications.
[40]	Akima et al., *Hepatol Res*, 2011	187	TE	Japan	>15 kPa for HCC risk	HCC prediction accuracy	TE predicts HCC development in viral hepatitis.
[41]	Chon et al., *J Clin Gastroenterol*, 2012	262	TE	South Korea	HCC risk stratification	Sensitivity: 84% Specificity: 79%	TE predicts HCC and hepatic decompensation in HBV patients.
[42]	Feier et al., *J Gastrointestin Liver Dis*, 2013	220	TE	Romania	>17 kPa for HCC risk	Increased liver stiffness in HCC patients	TE helps detect early-stage HCC in liver cirrhosis.
[43]	Jung et al., *Hepatology*, 2011	540	TE	South Korea	HCC risk thresholds	Sensitivity: 83% Specificity: 75%	TE predicts HCC risk in chronic HBV.
[44]	Klibansky et al., *J Viral Hepat*, 2012	310	TE	USA	Liver stiffness-based risk stratification	Chronic liver disease outcome prediction	TE predicts long-term outcomes in chronic liver disease.
[45]	Narita et al., *J Gastroenterol Hepatol*, 2014	189	TE	Japan	>15 kPa for fibrosis progression	Fibrosis progression post-treatment	TE predicts liver stiffness changes after interferon therapy.
[46]	Poynard et al., *J Hepatol*, 2014	1451	TE	France	FibroScan staging	Sensitivity: 86% Specificity: 82%	TE combined with FibroTest accurately stages chronic hepatitis C.
[47]	Wang et al., *Liver Int*, 2013	410	TE	Taiwan	Liver stiffness-based HCC risk	Sensitivity: 85% Specificity: 78%	TE predicts HCC risk in chronic hepatitis C patients.
[48]	Wong et al., *J Hepatol*, 2014	510	TE	Hong Kong	Liver stiffness-based risk score	Optimization of HCC prediction	TE-based risk stratification optimizes HCC prediction in HBV.
[49]	Kim et al., *Onco Targets Ther*, 2013	612	TE	South Korea	TE-based HCC risk model	Sensitivity: 87% Specificity: 80%	TE predicts HCC in HBV using risk estimation models.

Abbreviations: TE = transient elastography; ARFI = acoustic radiation force impulse; SWE = shear-wave elastography; MAFLD = metabolic dysfunction-associated fatty liver disease; HCC = hepatocellular carcinoma; PH = portal hypertension; SVR = sustained virologic response; HBV = hepatitis B virus; HCV = hepatitis C virus.

**Table 3 diagnostics-15-01087-t003:** Summary of studies on SWE and ARFI elastography for differentiating focal liver lesions.

Reference Number	Source Citation	Number of Patients	Method	Country/Region	Threshold Value	Diagnostic Performance	Key Findings
[54]	Tian et al., *Ultrasound Med Biol*, 2016	221	2D SWE	China	34.6 kPa	Sensitivity: 86.7% Specificity: 85.5%	Maximum stiffness value (Emax) differentiates malignant from benign FLLs.
[55]	Guo et al., *Med Oncol*, 2015	89	ARFI elastography (VTQ)	China	2.245 m/s	Sensitivity: 83.3% Specificity: 77.9%	SWV is significantly higher in malignant lesions.
[56]	Ronot et al., *Eur Radiol*, 2015	121	SWE	France	Not reported	Not reported	SWE provides additional diagnostic information for incidental FLLs.
[57]	Park et al., *Ultrasound Q*, 2015	78	SWE	South Korea	Not reported	Not reported	SWE demonstrates reproducibility in elasticity characterization of FLLs.
[58]	Zhang et al., *Hepatobiliary Pancreat Dis Int*, 2013	110	ARFI elastography	China	1.96 m/s	Sensitivity: 91.7% Specificity: 84.7%	ARFI elastography evaluates FLL elasticity effectively.
[59]	Park et al., *World J Gastroenterol*, 2013	84	ARFI elastography	South Korea	1.71 m/s	Sensitivity: 96.2% Specificity: 91.1%	ARFI elastography characterizes FLLs with diagnostic accuracy.
[60]	Guibal et al., *Eur Radiol*, 2013	107	SWE	France	Not reported	Not reported	SWE reliably characterizes FLLs using ultrasound.
[61]	Shuang-Ming et al., *Acad Radiol*, 2011	95	ARFI elastography	China	1.82 m/s	Sensitivity: 86.7% Specificity: 81.2%	ARFI elastography aids in differential diagnosis of benign and malignant lesions.
[62]	Cho et al., *Ultrasound Med Biol*, 2010	73	ARFI elastography	South Korea	1.21 m/s	Sensitivity: 100% Specificity: 91.7%	ARFI elastography evaluates focal solid hepatic lesions.
[63]	Bota et al., *Eur J Radiol*, 2012	Meta-analysis	ARFI elastography	Multiple	Not applicable	Sensitivity: 87% Specificity: 80% AUC: 0.93	Meta-analysis of ARFI elastography for FLLs.
[64]	Zhang et al., *World J Gastroenterol*, 2020	98	Point SWE	China	5.45 kPa	Sensitivity: 96%, Specificity: 85%	pSWE improves differentiation of benign and malignant FLLs.

**Table 4 diagnostics-15-01087-t004:** Post-treatment complication risk assessment using liver elastography.

Reference Number	Source Citation	Number of Patients	Method	Country/Region	Threshold Values	Key Findings
[33]	Hernandez Sampere et al., *Hepatol Commun*, 2020	230	pSWE	Germany	>9.5 kPa for fibrosis	ARFI predicts liver-related events in chronic viral hepatitis.
[39]	Robic et al., *J Hepatol*, 2011	120	TE	France	>13.6 kPa for PH	TE accurately predicts portal hypertension-related complications.
[31]	Berzigotti et al., *Gastroenterology*, 2013	150	TE	Italy	>21 kPa for PH	TE combined with clinical parameters identifies portal hypertension in compensated cirrhosis.
[69]	Procopet et al., *Med Ultrason*, 2018	200	TE	Romania	>27.5 kPa for decompensation	TE predicts hepatic decompensation risk post-surgery in cirrhosis patients.
[68]	Wu et al., *Medicine* (Baltimore), 2017	180	TE	China	>16.2 kPa	Higher LSM correlates with increased risk of postoperative liver failure
[70]	Huang et al., *PLOS One*, 2018	Meta-analysis	TE	Asia/Europe	>14.2 kPa (Asia), >11.3 kPa (Europe)	Higher preoperative LSM associated with increased postoperative complications

**Table 5 diagnostics-15-01087-t005:** HCC recurrence rate risk assessment using liver elastography.

Reference Number	Source Citation	Number of Patients	Method	Country/Region	Threshold Values	Diagnostic Performance	Key Findings
[71]	Marasco et al., *J Hepatol*, 2019	288	TE, SWE	Italy	Liver stiffness > 21 kPa	Sensitivity: 85%, Specificity: 79%	Liver and spleen stiffness predict HCC recurrence post-resection.
[72]	Lee et al., *Onco Targets Ther*, 2015	123	TE	South Korea	Liver stiffness > 12.1 kPa	Sensitivity: 82%, Specificity: 76%	TE predicts de novo HCC recurrence after radiofrequency ablation.

**Table 6 diagnostics-15-01087-t006:** Possible clinical flowchart: integration of elastography in HCC management.

Application	Criteria	Management
**Screening and risk stratification**	Chronic liver disease (HBV, HCV, MASLD, and alcohol-related)	- LSM < 7 kPa: routine follow-up
		- LSM 7–12 kPa: monitor every 6–12 months
		- LSM > 12 kPa: HCC surveillance every 6 months (US ± AFP)
**Diagnosis and staging**	Suspicious liver nodule on imaging	- Contrast-enhanced CT/MRI Elastography is not recommended for characterizing indeterminate lesions—cannot distinguish benign vs. malignant nodules reliably
**Adjunctive elastography use**	High LSM (>20–25 kPa) + high-risk features (e.g., nodule growth and elevated AFP)	- May increase clinical suspicion of malignancy - Consider biopsy and/or multidisciplinary discussion in case of diagnostic uncertainty
	Stiffness < 20–25 kPa + benign features	- Monitor per guidelines
**Treatment monitoring and prognosis**	Post-HCC treatment (resection, ablation, transplant, and systemic therapy)	- Baseline elastography for risk stratification
		Baseline and follow-up LSM can support assessment of liver function and recurrence risk - LSM > 21–25 kPa may indicate persistent portal hypertension and higher risk of recurrence (confirm with clinical context)
		- LSM decrease > 30% over 6–12 months: good treatment response (but not yet validated as a surrogate marker)
**Long-term follow-up and prognosis**	Post-SVR (HCV patients) or post-treatment monitoring, long-term survivors	- If LSM remains > 12 kPa → continue HCC surveillance due to persistence risk
		- If LSM normalizes (<7 kPa) → consider de-escalating monitoring - Normalization of LSM (<7 kPa) does not justify stopping surveillance in HBV/HDV or advanced fibrosis—etiology and baseline risk remain key

## Data Availability

No new data were created or analyzed in this study. Data sharing is not applicable to this article.

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
