# Peer review of "Liver Elastography Methods for Diagnosis of De Novo and Recurrent Hepatocellular Carcinoma"

_diagnostics, 2025, doi:10.3390/diagnostics15091087_

Round 1

Reviewer 1 Report

Comments and Suggestions for Authors

This is a well organized paper, but the authors’ analysis is not sufficient and is not accepted for publication.

Major points

1) (Ls.81-83): “Malignant tumors typically exhibit heightened cellularity -----, resulting in increased stiffness.”

The following relationships are not so straightforward, please analyze these points deeply and carefully;

 tumor-cellularity, cellularity-stiffness, (true) stiffness-shear wave value.

2)Transient elastography-Shear wave elastography: These methods are largely different. Please describe them separately as possible.

3) Operated frequencies (L.134, and Table 1): Description of probe frequency used in SWE is not correct. Please re-check it.

4) Description not directly related to liver elastography is too long (e.g. Ls172-181). Please shorten it as possible.

5) Threshold values: There is a large difference between previous reports. Please summarize these results and present some reasonable conclusions (the authors’ conclusion).

Minor points

1)     Ls 90-92: Static compression---. Please describe briefly this method. (It is considered to be non-diagnostic)

2)     Tumor characterization: “The characterization of liver lesions is outside the aim of this literature—“. However, the authors present a Table (Table 3). Please summarize the results.

Author Response

Major Points

1. Tumor Cellularity, Cellularity-Stiffness, and Stiffness-Shear Wave Value (Lines 81-83):
You raise a very important point regarding the relationship between tumor-cellularity, cellularity-stiffness, and stiffness-shear wave values. Indeed, the relationship between cellularity and stiffness is not as straightforward as initially suggested.

We have expanded the text: Malignant tumors often exhibit increased cellularity, which, alongside extracellular matrix alterations and stromal remodeling, can contribute to greater tissue stiffness. However, the relationship between cellularity and mechanical stiffness is complex and influenced by additional factors such as fibrosis, necrosis, and tumor heterogeneity. Consequently, tumor stiffness emerges as a potential imaging biomarker for distinguishing tumor characteristics, but it should be considered a complementary tool, not a stand-alone diagnostic criterion. This provides an opportunity for personalized care, as stiffness values can guide treatment decisions and monitor therapeutic response.

2. Transient Elastography vs. Shear Wave Elastography (Line 134 and Table 1):
We described both methods separately to clarify their differences and provide a more accurate understanding of each.  

Transient elastography

The initial commercially available TE system, the FibroScan from Echosens, was launched in 2003, in Europe and was approved by the Food and Drug Administration (FDA) in the United States in 2013. This system utilizes a 50-Hz mechanical impulse externally on the skin, followed by the measurement of the shear wave velocity. A range of probes is available, with the M probe primarily utilized for routine evaluations, whereas the XL probe enhances precision in patients with higher body mass or increased thoracic-abdominal wall thickness [[i]] Unlike the M probe, the XL probe measures deeper tissue layers (35–75 mm vs. 25–65 mm) and functions at a lower frequency (2.5 MHz vs. 3.5 MHz). ARFI elastography employs targeted ultrasound pulses to induce tissue displacement, leading to the propagation of shear waves for stiffness assessment [[ii]].

Acoustic Radiation Force Impulse (ARFI) based methods

ARFI (Acoustic Radiation Force Impulse) elastography is a non-invasive imaging technique used to assess tissue stiffness and works by applying short, focused ultrasonic pulses to tissue, generating localized displacements.

The terminology associated with ARFI elastography in scientific literature remains inconsistent. Although both pSWE and 2D SWE utilize ARFI to generate shear waves, current research often refers to pSWE specifically as ARFI elastography. Siemens (pSWE, Virtual Touch Quantification) and Supersonic Imagine (2D SWE) were among the first to introduce ARFI techniques in clinical applications. Today, these methods have been adopted by other leading manufacturers, including Philips [[iii]], Hitachi [[iv]], GE [[v]], Toshiba [[vi]] and Samsung [[vii]].

[i] Kennedy P, Wagner M, Castéra L, et al. Quantitative Elastography Methods in Liver Disease: Current Evidence and Future Directions. Radiology. 2018;286(3):738-763. doi:10.1148/radiol.2018170601

[ii] Gennisson JL, Deffieux T, Fink M, Tanter M. Ultrasound elastography: principles and techniques. Diagn Interv Imaging. 2013;94(5):487-495. doi:10.1016/j.diii.2013.01.022

[iii] Mare R, Sporea I, LupuÅŸoru R, et al. The value of ElastPQ for the evaluation of liver stiffness in patients with B and C chronic hepatopathies. Ultrasonics. 2017;77:144-151. doi:10.1016/j.ultras.2017.02.005

[iv] Dietrich CF, Dong Y. Shear wave elastography with a new reliability indicator. J Ultrason. 2016;16(66):281-287. doi:10.15557/JoU.2016.0028

[v] Sporea I, Bende F, Åžirli R, et al. The performance of 2D SWE.GE compared to transient elastography for the evaluation of liver stiffness. Ultraschall Med. 2016;37(S 01):SL19_3. doi: 10.1055/s-0036-1587806

[vi] Yang YP, Xu XH, Guo LH, et al. Qualitative and quantitative analysis with a novel shear wave speed imaging for differential diagnosis of breast lesions. Sci Rep. 2017;7:40964. Published 2017 Jan 19. doi:10.1038/srep40964

[vii] Piscaglia F, Salvatore V, Mulazzani L, et al. Differences in liver stiffness values obtained with new ultrasound elastography machines and Fibroscan: A comparative study [published correction appears in Dig Liver Dis. 2018 Jun;50(6):633. doi: 10.1016/j.dld.2018.03.015.]. Dig Liver Dis. 2017;49(7):802-808. doi:10.1016/j.dld.2017.03.001

3. Probe Frequency (Line 134 and Table 1):

We rechecked and revise the description of the probe frequencies, as the current information was inaccurate

Table 1: Overview of different elastography techniques

Technique

Ultrasound Frequency (MHz)

Availability

Cost

Liver-sampling volume

Region of Interest

Cause of Measurement Inaccuracy

Evidence

Transient  elastography

2.5-3.5 MHz

Widespread

Low

Small

Restricted, no guidance

Ascites, Obese patients

Excellent validation

Point Shear-Wave Elastography (pSWE)

4–9 MHz for deep tissue

9–15 MHz for superficial tissue

Moderate

Low

Small

Flexible with US guidance; recommended 1 cm below liver capsule and , 5 cm from transducer

High body mass index Standardization needed

Moderate,  good  validation

2D Shear-Wave Elastography (2D SWE)

Limited

Low

Medium

Flexible with US guidance

High body mass index Standardization needed

Moderate,  good  validation

4. Non-liver Elastography Content (Lines 172-181):
I acknowledge that the discussion is somewhat lengthy. We have shortened this section. 

Direct-acting antivirals (DAAs) have significantly improved HCV natural history, leading to high cure rates and reduced liver transplant needs. However, some patients still develop HCC post- sustained virological response (SVR), highlighting the importance of fibrosis assessment for prognosis and monitoring. Research indicates liver stiffness can decrease by up to 35% after DAA therapy, as measured by TE [[i]-[ii]] and acoustic radiation force impulse (ARFI) methods [[iii]]. Ongoing surveillance with elastography allows for early detection of stiffness increases, prompting further investigation and potentially influencing HCC screening strategies.

[i] Elsharkawy A, Alem SA, Fouad R, et al. Changes in liver stiffness measurements and fibrosis scores following sofosbuvir based treatment regimens without interferon. J Gastroenterol Hepatol. 2017;32(9):1624-1630. doi:10.1111/jgh.13758

[ii] Sáez-Royuela F, Linares P, Cervera LA, et al. Evaluation of advanced fibrosis measured by transient elastography after hepatitis C virus protease inhibitor-based triple therapy. Eur J Gastroenterol Hepatol. 2016;28(3):305-312. doi:10.1097/MEG.0000000000000533

[iii] Suda T, Okawa O, Masaoka R, et al. Shear wave elastography in hepatitis C patients before and after antiviral therapy. World J Hepatol. 2017;9(1):64-68. doi:10.4254/wjh.v9.i1.64

5. Threshold Values
The variation in threshold values reported in previous studies is indeed significant. We discussed more over this topic. 

While threshold values vary, likely due to differences in equipment and patient populations, the overall trend supports the diagnostic accuracy of elastography. Notably, point SWE (pSWE) shows particular promise in improving differentiation.

Importantly, our group was among the first to demonstrate the utility of real-time strain elastography in the noninvasive diagnosis of small HCC nodules in cirrhotic patients. In a prospective study of 42 patients with 58 liver nodules (1–3 cm), we found that color-coded strain patterns specifically increased blue color intensity were independently predictive of HCC, with a diagnostic accuracy (AUROC) of 0.94. The combination of real-time elastography and Doppler ultrasound significantly improved detection of small, sometimes hypovascular, HCCs lesions often missed by contrast-enhanced imaging. This approach was particularly useful in avoiding liver biopsy in high-risk or transplant-listed patients and provided valuable noninvasive confirmation of malignancy [[i]].

In essence, ultrasound elastography offers a noninvasive tool for improving the characterization of focal liver lesions. It has the potential to reduce the need for invasive procedures and guide clinical decision-making in HCC diagnosis and treatment planning. However, further research is still required to standardize cut-off thresholds, optimize acquisition protocols and integrate elastographic data into multiparametric diagnostic algorithms.

[i] Gheorghe L, Iacob S, Iacob R, et al. Real time elastography - a non-invasive diagnostic method of small hepatocellular carcinoma in cirrhosis. J Gastrointestin Liver Dis. 2009;18(4):439-446.

Minor Points

1. Static Compression Method (Lines 90-92): We described it briefly. 

  1. Static compression elastography (strain elastography) evaluates tissue stiffness by applying manual or physiological compression (natural organ activity such as cardiac motion, vascular pulsations, respiratory movements) and measuring tissue deformation. Unlike shear wave-based methods, it provides relative stiffness rather than absolute stiffness values. Due to operator dependency and lack of standardization, it is not recommended for liver fibrosis staging and is generally considered unsuitable for liver assessment [[i]].

[i] Dietrich CF, Barr RG, Farrokh A, Dighe M, Hocke M, Jenssen C, Dong Y, Saftoiu A, Havre RF. Strain Elastography - How To Do It? Ultrasound Int Open. 2017 Sep;3(4):E137-E149. doi: 10.1055/s-0043-119412. Epub 2017 Dec 7. PMID: 29226273; PMCID: PMC5720889.

Thank you for your valuable and insightful feedback. Your detailed comments have greatly helped refine and improve the manuscript. We sincerely appreciate the time and effort you have taken to provide such thoughtful suggestions.

We have carefully addressed all the points you raised and revised the manuscript accordingly.

The revised manuscript has been uploaded with all changes highlighted in yellow for easy reference. Please let us know if any further refinements are needed.

Reviewer 2 Report

Comments and Suggestions for Authors

Thanks for the opportunity to review this insightful and well-structured manuscript. Below are my detailed comments.

  • Consider adding a concise “Clinical Algorithm” or “Practical Clinical Flowchart” that shows how elastography can be integrated into standard HCC management pathways. This would help readers quickly translate the findings into practice.
  • The discussion on the limitations of elastography (e.g., body habitus, ascites, flare of transaminases, high BMI) is important. Including a brief section that outlines best practices (e.g., minimum fasting time, patient positioning, probe type, or recommended ROI) would strengthen the methods section for clinicians.
  • Highlight the lack of universal standardization across platforms. This problem can lead to confusion when comparing TE, pSWE, and 2D SWE from different vendors. Suggest future directions for unifying or standardizing measurements and reporting parameters.
  • Given the clinical importance of HCC surveillance, especially in cirrhotic patients, you could incorporate a section on how elastography (or changes in elastography readings over time) might complement current surveillance guidelines (e.g., ultrasound ± alpha-fetoprotein every 6 months).
  • Expand on how elastography and AI-integration might, in the future, replace or augment certain invasive or less-sensitive approaches to HCC detection.
  • The section on AI (artificial intelligence) and advanced imaging is particularly timely. Consider elaborating how AI tools might interpret elastographic data (e.g., automated ROI selection, real-time motion correction) and how these developments may further refine patient risk stratification.
  • You could also mention the potential integration of elastography with other emerging biomarkers (proteomic, metabolomic, or circulating tumor DNA) to enhance early detection of HCC.

Author Response

  • Consider adding a concise “Clinical Algorithm” or “Practical Clinical Flowchart” that shows how elastography can be integrated into standard HCC management pathways. This would help readers quickly translate the findings into practice.
  • The discussion on the limitations of elastography (e.g., body habitus, ascites, flare of transaminases, high BMI) is important. Including a brief section that outlines best practices (e.g., minimum fasting time, patient positioning, probe type, or recommended ROI) would strengthen the methods section for clinicians.
  • Highlight the lack of universal standardization across platforms. This problem can lead to confusion when comparing TE, pSWE, and 2D SWE from different vendors. Suggest future directions for unifying or standardizing measurements and reporting parameters.
  • Given the clinical importance of HCC surveillance, especially in cirrhotic patients, you could incorporate a section on how elastography (or changes in elastography readings over time) might complement current surveillance guidelines (e.g., ultrasound ± alpha-fetoprotein every 6 months).
  • Expand on how elastography and AI-integration might, in the future, replace or augment certain invasive or less-sensitive approaches to HCC detection.
  • The section on AI (artificial intelligence) and advanced imaging is particularly timely. Consider elaborating how AI tools might interpret elastographic data (e.g., automated ROI selection, real-time motion correction) and how these developments may further refine patient risk stratification.
  • You could also mention the potential integration of elastography with other emerging biomarkers (proteomic, metabolomic, or circulating tumor DNA) to enhance early detection of HCC.

Thank you for your thoughtful and insightful suggestions. We appreciate your recommendations to enhance the clarity, clinical applicability, and future directions. I have uploaded a modified version of the manuscript with all recommendations highlighted in yellow 

Round 2

Reviewer 1 Report

Comments and Suggestions for Authors

The manuscript has been largely revised, but it needs further revisions.

Minor points

1)Keywords: Please list some appropriate words, such as liver, ultrasound elastography, etc.

  2)P.3, Ls.94-100:-- Unlike shear wave-based methods,---is generally considered unsuitable for liver assessment.

    I agree with the authors in this opinion. For this reason, P.9, Ls 251-259/”importantly---confirmation of malignancy.” this paragraph is contradictory. Please elaborate on why it might be useful for HCC and other malignancies, or delete it.

 3) Please explain briefly the SMART-HCC score (P.16, L.363)

 4) Best practice for liver elastography in clinical use. It describes transient elastography only, please add that of SWE.

Author Response

Dear Reviewer,

We thank you for your valuable comment and the opportunity to clarify and expand on this point - utility of strain elastography.

We fully agree that strain elastography (static compression elastography) has clear limitations in the evaluation of diffuse liver disease and is not recommended for fibrosis staging due to its operator dependency and lack of standardization. However, we would like to emphasize that, beyond fibrosis assessment, several studies — including our own — have explored the potential utility of real-time strain elastography (RTE) in focal liver lesions, particularly hepatocellular carcinoma (HCC).

In fact, our group was among the first to demonstrate that real-time strain elastography may provide complementary information in the noninvasive characterization of small HCC nodules in cirrhotic patients, with high diagnostic accuracy (AUROC 0.94), particularly when combined with Doppler ultrasound. Although this approach has not become widely implemented in routine clinical practice, likely due to technical limitations and the emergence of more standardized methods such as shear wave elastography (SWE), the concept has remained of interest.

More recent studies have continued to investigate the role of RTE, especially with the introduction of new convex probes with better penetration and improved software allowing semiquantitative assessment. As highlighted in a review by Sandulescu et al. (J Gastrointestin Liver Dis, 2013), these technical improvements renewed interest in RTE for liver applications, including tumor characterization and ablation monitoring.

Moreover, a recent experimental study by Li et al. (Diagnostics 2023) demonstrated the potential of RTE, alongside SWE, in evaluating the immediate effects of radiofrequency ablation in liver tissue. Their results confirmed that both methods could accurately delineate ablation zones, with good agreement between RTE, SWE, and histological measurements.

Although these applications are still in the research field and not yet standard of care, we believe they are relevant to mention, especially in the context of liver oncology, where noninvasive tools for lesion characterization and treatment monitoring are evolving rapidly.

We have now added a dedicated paragraph in the discussion to reflect this point and to acknowledge both the limitations and the potential niche applications of RTE in liver malignancies.

We have also addressed all the other points